# Multi-Mycotoxin Analysis in Italian Grains Using Ultra-High-Performance Chromatography Coupled to Quadrupole Orbitrap Mass Spectrometry

**DOI:** 10.3390/toxins15090562

**Published:** 2023-09-08

**Authors:** Juliane Lima da Silva, Sonia Lombardi, Luigi Castaldo, Elena Morelli, Jaqueline Garda-Buffon, Luana Izzo, Alberto Ritieni

**Affiliations:** 1School of Chemistry and Food, Federal University of Rio Grande, Av. Itália, Km 8, Rio Grande 96203-900, RS, Brazil; julianelima@furg.br (J.L.d.S.); jaquelinebuffon@furg.br (J.G.-B.); 2Department of Pharmacy, University of Naples Federico II, Via Domenico Montesano 49, 80131 Naples, Italy; luigi.castaldo2@unina.it (L.C.); elena.morelli@unina.it (E.M.)

**Keywords:** grain samples, mycotoxins, UHPLC Q-Orbitrap HRMS, co-occurrence

## Abstract

Mycotoxins are a major source of contamination in cereals, posing risks to human health and causing significant economic losses to the industry. A comprehensive strategy for the analysis of 21 mycotoxins in Italian cereal grain samples (*n* = 200) was developed using a simple and quick sample preparation method combined with ultra-high-performance liquid chromatography coupled with quadrupole Orbitrap high-resolution mass spectrometry (UHPLC Q-Orbitrap HRMS). The proposed method showed some advantages, such as multi-mycotoxin analyses with simple sample preparation, fast determination, and high sensitivity. The analysis of the sample revealed the presence of 11 mycotoxins, with α-zearalenol being the most frequently detected, while deoxynivalenol exhibited the highest contamination level. Furthermore, co-occurrence was identified in 15.5% of the samples under analysis. Among these, 13% of the samples reported the simultaneous presence of two mycotoxins, while 2.5% showed the co-occurrence of three mycotoxins. Currently, there has been a renewed interest in guaranteeing the quality and safety of products intended for human consumption. This study holds significant value due to its ability to simultaneously detect multiple mycotoxins within a complex matrix. Furthermore, it provides findings regarding the occurrence and co-occurrence of emerging mycotoxins that currently lack regulation under the existing European Commission Regulation.

## 1. Introduction

Cereal grains stand as crucial food commodities on a global scale, assuming a significant role in human nutrition [1]. For thousands of years, cereals were a primary source of nourishment for humans. Nowadays, cereals represent the most important source of calories relative to a large part of the world’s population. Compared to the industrialized world, developing nations rely more on cereal grains to meet their nutritional demands. Several factors, such as water, environment, temperature, and economic and cultural availability, have an impact on grown crops and the typology of produced grains [2]. Wheat exhibits remarkable adaptability to diverse growth conditions, resulting in their widespread cultivation across the globe. According to the latest FAO data, the world’s annual production of wheat and products amounts to 761 million tonnes. The most substantial contributor is Asia, boasting an annual output of 347 million tonnes (45.6%), followed by Europe with wheat production of about 255 million tonnes (33.6%) [3]. The cultivation and proper management of these crops hold pivotal importance for ensuring global food security. Throughout the production process, cereal grains encounter various sources of fungal and bacterial contamination, potentially carrying a vast and varied microbial population [4,5]. In the context of fungal contamination, it is pertinent to acknowledge that the repercussions extend beyond mere degradation with respect to grain quality and nutritional attributes. A salient facet of such contamination lies in the potential biosynthesis of mycotoxins, which constitute natural contaminants that are indigenous to food matrices. Mycotoxins, recognized as secondary metabolites, emerge as products of fungal biosynthesis within the taxonomic realms of *Penicillium*, *Fusarium*, *Aspergillus*, and *Alternaria* genera. The propensity for mycotoxin contamination is manifested during each food production phase, encompassing the pre-harvest, post-harvest, processing, storage, and distribution phases [6]. This widespread presence highlights the interplay between the metabolic pathways of fungi and the complex network of the food supply chain, thereby substantiating the manifold dimensions of mycotoxin propagation. Mycotoxin occurrence is primarily influenced by factors that include the levels of carbon dioxide and oxygen, substrate composition, harvest maturity, temperature, pH, water activity, and the presence or absence of fungicides and pesticides [7]. The most commonly associated mycotoxins with respect to the pre-harvest phase are produced by *Fusarium* fungal species, including trichothecenes (deoxynivalenol and T-2 toxin), zearalenone, and fumonisins [5]. On the other hand, in the post-harvest phase, fungal species *Aspergillus* and *Penicillium* are more common and, depending on the conditions, may produce aflatoxins and ochratoxins [8]. These mycotoxins, in turn, hold the capacity to evoke a spectrum of deleterious effects in both animal and human cohorts. Notably, these effects encompass a range of toxicological manifestations, including teratogenicity, mutagenicity, carcinogenicity, immunotoxicity, and estrogenic damage [7,9]. The symbiotic interplay between fungal contamination and mycotoxin production underscores the gravity of their combined impact on both the agricultural and public health domains. Consequently, beyond the overt diminution of grain integrity, the intricate linkages between fungal-induced mycotoxin synthesis and multifaceted toxicological sequelae underscore the imperative for sustained vigilance in both agricultural practices and food safety protocols. Based on their carcinogenic capacity, certain mycotoxins, such as aflatoxins, have been categorized among human carcinogens by the International Agency for Research on Cancer (IARC) [10].

Contaminants can make their way into the human food chain either by the direct consumption of contaminated food or indirectly via residues present in eggs, meat, milk, and dairy products originating from initially contaminated animal feed [11,12]. It is approximated that around 25–50% of global cereal products have mycotoxin contamination, with 5 to 10% enduring irreversible contamination, resulting in substantial economic losses [13]. Consequently, there has been an increase in interest in creating a quick, high-throughput, and highly sensitive method to analyze multiple mycotoxin residues. Given that grain constitutes an intricate matrix, proper sample preparation is essential. To date, several methods for extracting mycotoxins using sample preparation have been developed. Among these, the QuEChERS method (quick, easy, cheap, effective, rugged, and safe) has showcased a wide range of advantages, alongside techniques such as liquid-liquid or solid-phase extraction and the salting out technique. These methods enable rapidity and the utilization of reduced volumes of organic solvents, and they achieve satisfactory recovery rates [14].

Methods for detecting mycotoxins include enzyme-linked immunosorbent assays (ELISAs), thin-layer chromatography (TLC), high-performance liquid chromatography (HPLC), gas chromatography coupled with mass spectrometry (GC-MS), and HPLC coupled with tandem mass spectrometry (HPLC-MS/MS) [15,16,17]. Although all these methods have known advantages, they also have substantial disadvantages. For instance, HPLC analysis exhibits reduced separation capacity and analysis speed, and HPLC-MS/MS encounters limitations in analyzing trace levels within complex matrices. These limitations are further pronounced when addressing the detection of multiple mycotoxins within complex food samples using any of the aforementioned methods. In that regard, because of the mass accuracy offered by a high-resolution mass spectrometry (HRMS) detector in combination with conventional data, HRMS using Orbitrap technology has been highlighted for achieving high-resolution results and good specificity. Orbitrap provides advantages that cover the disadvantages of traditional methods used precisely because of the high resolution, and it is able to determine molecule weights and their fragments in high resolution even in complex matrices due to their high anti-interference ability. In addition, unlike MS/MS, this method allows for retrospective data analysis, obviating the necessity to re-run samples [18,19]. Given the limited existing literature reporting the presence of these harmful compounds in Italian grain, the current study aimed to (i) optimize a simple multi-approach utilizing a salting out technique for mycotoxin extraction from grain samples, followed by quantification using the UHPLC-Q-Orbitrap HRMS methodology (ii) to assess the occurrence of mycotoxins (*n* = 21) in 200 Italian wheat grain samples.

## 2. Results

### 2.1. Optimization of Sample Preparation

Effective sample preparation stands as a pivotal phase in multi-residual protocols given the diverse physicochemical attributes of the analytes and the complex nature of the matrix. During the optimization process, three similar procedures were evaluated. In particular, critical extraction parameters, namely, extractor solvents and the clean-up stage, were obtained. Two extractor solvents (AcN treated with 0.1 and 5% formic acid) and the necessity of the clean-up step using C_18_ were tested and carried out. The efficiency of the methods was compared by assessing parameters such as recovery and matrix effects. The recovery of mycotoxins using 5% acidified acetonitrile showed better results compared to 0.1%. The effect matrix, i.e., the percentage of signal suppression or the enhancement effect outside the 80–120% range, was observed for some mycotoxins in the studied protocols. In addition, the results of the protocol tested without C_18_ (Figure 1: 1° protocol) showed better recovery results. Therefore, the protocol using 5% acidified acetonitrile without the clean-up step was settled.

### 2.2. Analytical Features of the Proposed Approach

The performance of the optimized method is summarized in Table 1. Calibration and matrix-matched curves were established for each analyte at eight concentration tiers, spanning from the limit of quantification (LOQ) to 400 µg/kg, with each curve prepared in triplicate. Good regression coefficients (>0.988) were achieved for all investigated mycotoxins. In appraising the matrix effect, matrix-matched calibration curves (A) were juxtaposed against standard calibration curves (B), with their relationship quantified as a percentage ratio of these slopes [(A/B) × 100], denoting the matrix effect (signal suppression enhancement, SSE, expressed in %). When SSE is lower than 100%, there is signal suppression; in contrast, when SSE is higher than 100%, there is signal enhancement. We obtained matrix effects ranging between 63 and 122%, which were considered in the calculation of the results. To ascertain the absence of potential interferences at the identical retention time as the targeted compounds, a blank grain sample underwent analysis. Trueness, expressed as recovery data, was calculated across a spectrum spanning from 2 to 500 µg/kg depending on the analytes (Table 1).

This method provided recoveries ranging between 74 and 134%. The method’s precision was assessed through the lens of both repeatability (intra-day precision, *n* = 4) and reproducibility (inter-day precision, *n* = 4), utilizing grain samples spiked at the aforementioned levels. Based on the limits set out in the regulation [20], for DON and FUMs, spike levels of 50–500 µg/kg were tested. Regarding AFs, spike levels of 2–100 µg/kg were evaluated.

The precision results, denoted as the relative standard deviation (RSD), revealed the commendable repeatability and reproducibility of the suggested approach. Additionally, the limits of detection (LOD) and LOQ were established for each respective target compound. The determination of the LOD was based on the minimum tested concentration, enabling the identification of the molecular ion with a mass error of <5 µg/kg. The LOQ was ascertained as the minimum analyte concentration that yielded a chromatographic peak, exhibiting accuracy and precision within acceptable parameters (<20%). The calculated LOQ spanned from 0.39 to 6.25 µg/kg.

### 2.3. Real Samples Analyses

A total of eleven distinct mycotoxins were identified within the examined samples of Italian grain (*n* = 200). Notably, out of the entire sample set, contamination by at least one mycotoxin was reported in 116 samples. ZEN metabolites, α-ZAL and β-ZAL, were the most frequently detected compounds, and they were detected in 42% and 8% of samples, respectively, indicating that wheat can be easily contaminated with ZEN and, consequently, those metabolites. The α-ZAL concentration range detected was from 19.58 to 147.20 µg/kg, while the β-ZAL concentration was between 31.48 and 176.88 µg/kg. Moreover, HT-2 was identified in a substantial 65% of the examined grain samples, with concentrations ranging from 3.3 to 28.34 µg/kg. Additionally, β-ZEL was identified in four samples, exhibiting concentrations ranging from 6.52 to 12.44 µg/kg, while DON was quantified in three samples in the levels between 104.08 and 292.62 µg/kg. The presence of ZAN was observed in two samples, with 4.58 and 4.88 µg/kg of contamination. Furthermore, α-ZEL, T-2 toxin, ENN A1, ENN B1, and ENN A were detected in just one sample (Table 2).

Despite the detection of mycotoxins in 58% of the assayed samples, none of them exceeded the limit recommended by regulatory agencies [20]. The decision-making process for establishing the maximum levels of mycotoxins in food has involved many factors, including scientific data regarding their occurrence and their toxicological aspects [21,22]. The Codex Alimentarius Commission, FAO, and WHO have the responsibility to impose acceptable limits for the harmonization of limits worldwide. Several countries have established regulations relating to mycotoxins in foods in order to guarantee safe consumption. The regulations are specifically related to mycotoxins AFs, OTA, trichothecenes, FBs, patulin, agaric acid, ergot alkaloids, sterigmatocystin, zearalenone, and phomopsins. In particular, regulatory and scientific interest in mycotoxins has increased in the European Union, in which harmonized limits exist [20,23,24]. DON, the mycotoxin exhibiting the highest detected concentration and one of greatest concern among the mycotoxins detected, was well below the established European Commission’s maximum threshold of 1750 µg/kg for unprocessed wheat [20]. This result shows that even though DON contamination has been detected, the levels do not appear to be dangerous (292.62 µg/kg). Furthermore, DON being detected in only two samples means that it is not a recurring contamination. Juan et al. [25] and Alkadri et al. [26] also detected DON with the highest levels of contamination in the Italian wheat that they analyzed, but they observed substantially higher concentrations than what we found: 2267 and 1230 µg/kg, respectively. Wu et al. [16] analyzed 63 mycotoxins in 63 wheat samples, and 100% of the samples showed contamination with at least one analyte. The toxins most frequently detected by the authors were tentoxin and deoxynivalenol 3-glucoside, which are compounds that are not evaluated by us. However, a wide occurrence of ENNB and ENNB1 was found in the analyzed samples, which corroborates that enniatins are mycotoxins that can easily contaminate wheat.

The co-occurrence of mycotoxins was reported in 15.5% of analyzed samples. Two mycotoxins were found in 13% of the samples, while three were found in 2.5%. The occurrence of multiple mycotoxins could impact their potential toxicity, as synergistic or additive effects have been previously noted in in vitro assessments [27]. The investigation into mycotoxin co-occurrence conducted by Smith et al. [28] unveiled prevalent combinations, including AFs + OTA, DON + ZEN, DON + NIV, and DON + T-2 toxins. In a study conducted by Juan et al. [29], noteworthy co-occurrences between AFs + FUM, DON + ZEN, AFs + OTA, and FUM + ZEN were observed the most. In this way, it is possible to identify that the combinations between aflatoxins and OTA or *Fusarium* toxins, such as fumonisins and ZEA, more commonly occur. Even though our co-occurrence results were different, we did not detect those mycotoxins alone in our study, and the combinations were between ZEN metabolites (α-ZAL, β-ZAL, and β-ZEL) and other toxins from *Fusarium* genera (ENNA1, T-2 toxin, DON, and HT-2), which is in accordance with the cited studies (Table 3).

As much as the levels detected in this study were not high and likely to be dangerous for human consumption, analytical studies with the aim of developing increasingly effective, simple, and green methods must be recurrent [30,31]. Occurrence studies in grains should be frequently carried out because contamination by mycotoxins is dependent on fungal stress and the various environmental factors of pre- and post-harvest phases that are already mentioned in this manuscript. Additionally, the values of the contamination must vary according to the crop and the wheat-planting region [32,33]. Since this is a matrix that is very well consumed by humans and animals, it is important to have control over the levels of these contaminants because they may present a serious health risk.

## 3. Conclusions

Mycotoxins pose a substantial risk of contamination in cereal grains, presenting a severe threat to human health and ranking as a paramount concern within the industry. Legislation calls for monitoring methods. The possibility of the rapid identification and control of the entire supply chain would be a real advantage for human health. In fact, it would prevent the unsafe consumption of already contaminated foods that are dangerous for humans. The purpose of this investigation was to develop a method for the simultaneous detection of 21 mycotoxins in grains and to apply it to 200 Italian grain samples. The method was demonstrated to be efficient and advantageous due to its simple preparation, rapid determination, and high sensitivity and capability in detecting a wide range of mycotoxins in grain samples. The analysis of samples confirmed the occurrence of up to 11 mycotoxins, with α-zearalenol being the most often found and deoxynivalenol showing the greatest amount of contamination. Additionally, co-occurrence was found in 15.5% of samples. Despite the contamination of grains by mycotoxins, it does not pose a threat to public health and respects the established regulatory limits.

## 4. Materials and Methods

### 4.1. Chemicals and Reagents

Methanol (MeOH), acetonitrile (AcN), formic acid (FA), and H_2_O for chromatography were acquired from Merck (Milan, Italy). Sodium chloride (NaCl), ammonium formate (NH_4_HCO_2_), anhydrous sulfate sodium (Na_2_SO_4_), sodium acetate (C_2_H_3_O_2_Na), anhydrous sodium acetate (NaAc), and discovery octadecyl silica sorbent (C_18_, analytical grade) were purchased from Sigma-Aldrich (Milan, Italy). Chemicals and reagents were of analytical grade and used for UHPLC–MS analyses.

Mycotoxin standards and metabolites (purity ≥ 98%), called aflatoxins (AFB_1_, AFB_2_, AFG_1_, and AFG_2_), ochratoxin A (OTA), fumonisins (FB_1_ and FB_2_), deoxynivalenol (DON), neosolaniol (NEO), HT-2 toxin, T-2 toxin, α-zearalanol (α-ZAL), α-zearalenol (α-ZEL), β-zearalanol (β-ZAL), β-zearalenol (β-ZEL), zearalanone (ZAN), zearalenone (ZEN), and enniatins (ENNB, ENNB1, ENNA, and ENNA1) were provided by Sigma Aldrich (Milan, Italy). For the preparation of individual stock solutions, each mycotoxin was diluted in methanol to reach a concentration of 1 mg/mL. A working standard solution including all the mycotoxins was obtained by diluting the stock solution in MeOH:H_2_O (70:30 *v*/*v*, 0.1% formic acid) to reach the concentrations needed for the spiking experiments (500, 20, and 2 µg/mL). The analytical standards were saved in a closed tightly container under cool dry conditions at −20 °C in a well-ventilated place, as stated in the safety data sheets reported by the manufacturer (Sigma Aldrich; Milan, Italy).

### 4.2. Sampling

Two hundred Italian grain samples grown in different fields situated in Campania, Italy’s southern region, were provided by farmers. All samples were kept in their original packages and maintained at 4 °C until the analysis, which was carried out within one week after their arrival in the laboratory.

### 4.3. Mycotoxin Extraction

Briefly, 2.5 g of homogenous grain sample was weighed and placed in a falcon tube (50 mL)m and 5 mL of UHPLC grade water was added. The obtained mixture was vortexed for 1 min, and then 5 mL of acidified AcN with 5% formic acid was added. After 30 s of vortexing and 2 min of horizontal shaking, MgSO4 (2 g) and NaCl (1 g) were added. Afterward, the falcon tube was manually agitated (1 min), vortexed (30 s), shaken (2 min), and centrifugated (5 min, 5000 rpm). The supernatant was filtered through a nylon filter (0.22 µm) and analyzed using HPLC-Q-Orbitrap HRMS [34].

### 4.4. UHPLC Q-Orbitrap HRMS Analysis

Mycotoxin identification was performed using a methodology previously optimized by Izzo et al. [35]. A UHPLC system (Dionex UltiMate 3000, Thermo Fisher Scientific, Waltham, MA, USA) composed of a micro-degasser system (GPL-3400RS), a thermostatic column oven (TCC-3000SD), a solvent delivery pump (LPG-3400RS), and a refrigerated autosampler (WPS-3000RS) was used for analyses. A Kinetex 2.6 μm column (100 × 2.1 mm, Phenomenex, Torrance, CA, USA) with a pre-column (5 × 2 mm, size of the particles 1.8 μm) was used for chromatography separation. The injection volume was set at 5 μL, and the flow rate was set at 500 uL/min. The mobile phases were water (A) and methanol (B) both contained formic acid (0.1%) and ammonium formate (5 mM). The sep aration conditions were as follows: The linear gradient for B started at 10% for 0.5 min and increased within 2.5 min to 80%; then, it increased to 100% at 3 min and then decreased to 10% in 2 min. The column was equilibrated at 10% for 1.5 min before the next injection. The total runtime was 9 min.

Mass spectrometry analysis was carried out using the electrospray ionization interface (ESI) in both the negative and positive ion modes. Full-scan and fragmentation spectrums in the all independent ion (AIFs) mode were analyzed. In the full-scan mode, the following scan parameters were set: mass resolving power of 35,000 FWHM; automatic gain control of 1 × 10^6^; time of maximum injection of 200 ms and scan rate of 3 scan/s; spray voltage of 2.8 kV; scan range 80–1200 *m*/*z*; sheath gas flow rate: 40 arbitrary unit (a.u.); auxiliary gas flow rate: 10 a.u.; heater temperature: 305 °C; capillary temperature: 310 °C; capillary voltage: 50 V; tube lens voltage: 110 V. In the AIF mode, the following scan parameters were set: 17,500 FWHM mass resolving power, 1 × 10^5^ for automatic gain control; 80–1200 *m*/*z* of the scan range; 200 ms maximum injection time; isolation window up to 5 *m*/*z*.

### 4.5. Method Validation

The method was optimized in accordance with the European regulations [36,37,38]. Matrix effect, specificity, linearity, trueness, precision, and sensitivity were evaluated. Data quality was ensured using a comprehensive range of quality control procedures. The retention time of analytes was compared with the standards, and a tolerance of ±2.5% was set for confirmation criteria. A rigorous and systematic control was included in each batch of samples. A blank reagent, procedural and replicate samples, and a matrix-matched calibration were analyzed.

### 4.6. Statistical Analysis

The results of the sample’s analysis, which was carried out in triplicate, were reported as mean ± RSD. Info-Stat 2008 was used to carry out the statistical analysis of the data. The level of *p* ≤ 0.05 was statistically significant.

## Figures and Tables

**Figure 1 toxins-15-00562-f001:**
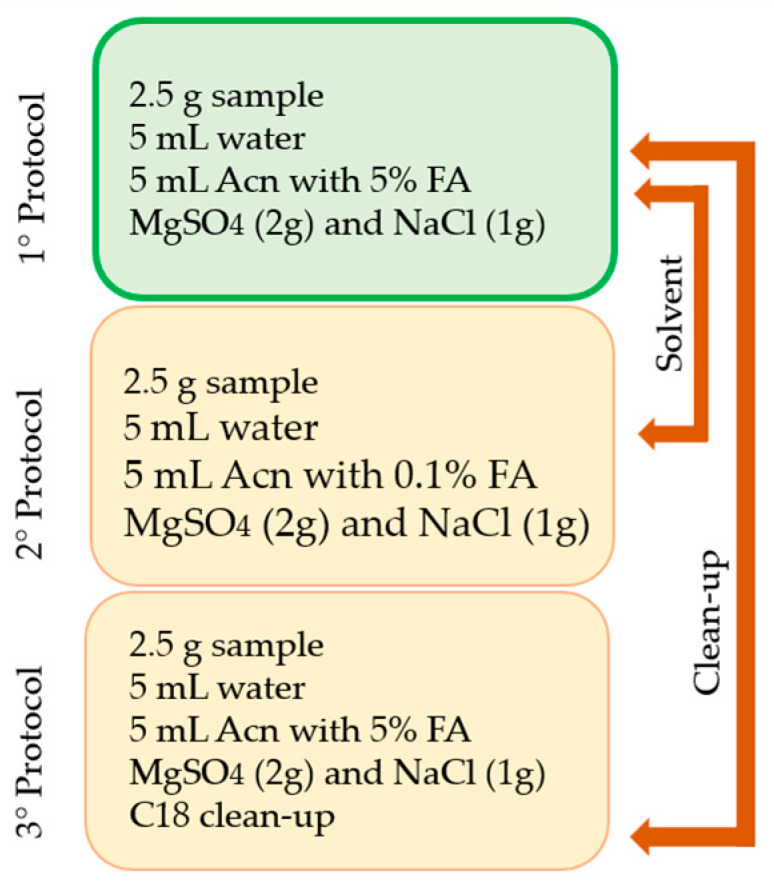
Comparison among the tested extraction protocols.

**Table 1 toxins-15-00562-t001:** Assessment of the method performance matrix effect (SSE %), linearity, recovery (values indicate spiked levels), and LOQ.

Analyte	Linearity (R2)	SSE (%)	Recovery (%) (RSD (%))				LOQ
			50 µg/kg	100 µg/kg	200 µg/kg	500 µg/kg	(µg/kg)
**DON**	0.988	101	98 (5)	115 (4)	75 (20)	89 (18)	6.25
**FB_1_**	0.996	93	114 (9)	75 (15)	114 (12)	107 (19)	6.25
**FB_2_**	0.998	86	101 (11)	102 (10)	114 (9)	104 (13)	0.78
			**2** **µg/kg**	**10** **µg/kg**	**50** **µg/kg**	**100** **µg/kg**	
**AFB_1_**	0.999	67	110 (8)	88 (12)	89 (8)	102 (6)	1.56
**AFB_2_**	1.000	80	85 (15)	108 (13)	124 (7)	121 (13)	0.78
**AFG_1_**	1.000	84	96 (11)	149 (10)	110 (11)	104 (7)	0.78
**AFG_2_**	0.998	86	78 (6)	122 (9)	108 (17)	102 (8)	3.12
			**5** **µg/kg**	**10** **µg/kg**	**50** **µg/kg**	**100** **µg/kg**	
**NEO**	1.000	70	75 (19)	87 (9)	96 (16)	94 (10)	1.56
**α-ZAL**	1.000	75	85 (11)	125 (11)	94 (9)	99 (8)	3.12
**β-ZAL**	0.998	81	96 (8)	106 (15)	103 (14)	96 (10)	1.56
**α-ZEL**	0.999	63	87 (12)	77 (10)	78 (16)	87 (8)	6.25
**β-ZEL**	0.996	80	84 (5)	102 (12)	102 (6)	98 (12)	6.25
**T_2_**	1.000	84	83 (9)	89 (6)	83 (8)	84 (11)	1.56
**HT-2**	1.000	71	87 (8)	134 (6)	116 (11)	105 (8)	1.56
**ENNA**	1.000	96	71 (6)	74 (14)	101 (13)	101 (13)	3.12
**ENNA_1_**	1.000	81	78 (4)	112 (7)	107 (15)	102 (15)	6.25
**ENNB**	1.000	122	101 (13)	116 (14)	99 (9)	103 (13)	3.12
**ENNB_1_**	0.998	79	89 (9)	80 (16)	111 (17)	98 (8)	3.12
**OTA**	0.999	68	99 (18)	94 (16)	108 (7)	98 (7)	3.12
**ZAN**	1.000	91	85 (14)	77 (8)	98 (12)	107 (10)	0.39
**ZEN**	0.999	88	71 (8)	96 (17)	105 (4)	105 (14)	1.56

Abbreviations: DON: deoxynivalenol; FB_1_: fumonisin B_1_; FB_2_: fumonisin B_2_; NEO: neosolaniol; α-ZAL: α-zearalanol; β-ZAL: β-zearalanol; α-ZEL: α-zearalenol; β-ZEL: β-zearalenol; T_2_: T-2 toxin, HT-2: toxin HT-2; ENNA: enniatin A; ENNA_1_: enniatin A_1_; ENNB: enniatin B; ENNB_1_: enniatin B_1_; OTA: ochratoxin A; AFB_1_: aflatoxin B_1_; AFB_2_: aflatoxin B_2_; AFG_1_: aflatoxin G_1_; AFG_2_: aflatoxin G_2_; ZAN: zearalanone; ZEN: zearalenone.

**Table 2 toxins-15-00562-t002:** Mycotoxins occurrence in Italian grain samples.

Analyte	Positive Samples	Percentage (%)	Concentration Range (µg/kg)
DON	3/200	1.5	104.08–292.62
α-ZAL	84/200	42	19.58–147.20
β-ZAL	28/200	14	31.48–176.88
α-ZEL	1/200	0.5	13.99
β-ZEL	4/200	2	6.52–12.44
T_2_	1/200	0.5	2.76
HT-2	9/200	4.5	3.3–28.34
ENNA	1/200	0.5	1.28
ENNA_1_	1/200	0.5	2.62
ENNB_1_	1/200	0.5	2.18
ZAN	2/200	1	4.58; 4.88

Abbreviations: DON: deoxynivalenol; α-ZAL: α-zearalanol; β-ZAL: β-zearalanol; α-ZEL: α-zearalenol; β-ZEL: β-zearalenol; T_2_: T-2 toxin, HT-2: toxin HT-2; ENNA: enniatin A; ENNA_1_: enniatin A_1_; ENNB_1_: enniatin B_1_; ZAN: zearalanone.

**Table 3 toxins-15-00562-t003:** Combinations of mycotoxins and the frequency and sum of obtained concentrations.

Co-Occurrence Combination	Combination Frequency	∑ Concentration (µg/kg)
β-ZAL, ENN A1	1	29.48
β-ZAL, β-ZEL	1	51.15
β-ZAL, α-ZAL	11	903.19
T-2, α-ZAL	1	31.14
DON, α-ZAL	1	195.53
α-ZEL, β-ZAL	1	32.30
HT-2, α-ZAL	2	61.10
HT-2, α-ZAL, β-ZAL	1	87.48

Abbreviations: α-ZAL: α-zearalanol; β-ZAL: β-zearalanol; β-ZEL: β-zearalenol; ENNA_1_: enniatin A_1_; T_2_: T-2 toxin; DON: deoxynivalenol; HT-2: toxin HT-2; ZAN: zearalanone.

## Data Availability

Not applicable.

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
