# Peer review of "Multi-Mycotoxin Analysis in Italian Grains Using Ultra-High-Performance Chromatography Coupled to Quadrupole Orbitrap Mass Spectrometry"

_toxins, 2023, doi:10.3390/toxins15090562_

Round 1
Reviewer 1 Report
This manuscript describes the detection and analysis of multi-mycotoxins in Italian cereals using UPLC-Q-Orbitrap-MS. This study is of great value for the risk assessment of common and emerging mycotoxin contamination in grains in Europe. While the paper is well-prepared and organized, a few minor revisions are suggested to enhance its elegance and readability.
1. Line 57 and thereafter, “T2” should be “T-2 toxin”.
2. Line 79, there are two periods after the word “essential”.
3. Line 90-91, the author said that “HPLC-MS/MS encounters limitations in analyzing trace levels within complex matrices.”, however, other researchers have already developed several methods for multi-mycotoxin detection in cereal matrix, and their contributions should be cited here.
4. Section 2.1 & Figure 1, the authors evaluated three procedures, however, the figure 1 is quite confusing, as Protocol 1 & 4 were same. So what is the point of this figure?
5. Line 158, please add the exact sample size (n=200) to make it clear.
6. Line 171, please clarify the regulatory agencies.
7. Table 3, it is better if the authors add the frequencies of each co-occurrence combination.
Author Response
Manuscript ID: toxins-2573433
Title: Multi-mycotoxins analysis in Italian grains by using ultra-high-performance chromatography coupled to quadrupole orbitrap mass spectrometry
Response to Reviewer 1
Comments and Suggestions for Authors
This manuscript describes the detection and analysis of multi-mycotoxins in Italian cereals using UPLC-Q-Orbitrap-MS. This study is of great value for the risk assessment of common and emerging mycotoxin contamination in grains in Europe. While the paper is well-prepared and organized, a few minor revisions are suggested to enhance its elegance and readability.
Point 1: Line 57 and thereafter, “T2” should be “T-2 toxin”.
Response 1: As suggested by reviewer 1, the authors changed T2 as T-2 toxin.
Point 2: Line 79, there are two periods after the word “essential”.
Response 2: As suggested by reviewer 1, the authors deleted the period.
Point 3: Line 90-91, the author said that “HPLC-MS/MS encounters limitations in analyzing trace levels within complex matrices.”, however, other researchers have already developed several methods for multi-mycotoxin detection in cereal matrix, and their contributions should be cited here.
Response 3: As suggested by reviewer 1, the authors added appropriately references.
Point 4: Section 2.1 & Figure 1, the authors evaluated three procedures, however, the figure 1 is quite confusing, as Protocol 1 & 4 were same. So what is the point of this figure?
Response 4: As suggested by reviewer 1, the authors changed the figure 1.
Point 5: Line 158, please add the exact sample size (n=200) to make it clear.
Response 5: As suggested by reviewer 1, the authors added the number if samples in line 158.
Point 6: Line 171, please clarify the regulatory agencies.
Response 6: As suggested by reviewer 1, the authors added the regulatory agencies.
Point 7: Table 3, it is better if the authors add the frequencies of each co-occurrence combination.
Response 7: As suggested by reviewer 1, the authors added the combination frequency.
Thank you for the time spent evaluating our manuscript.
Reviewer 2 Report
The authors present an important study for complex multi-toxin analysis. The study is technically state of the art and the manuscript is well written. In the Discussion additional information is necessary for the readers: Please state what steps had so far to be carried out for comparable results using standard techniques - in comparison to the results using the present combination of ultra high-performance liquid chormatorgraphy together with high-resolution mass spectrometry. Is it possible to give an overview of the resources/costs of the so far methods and the innovative multi-sample approach - with respect to instruments/materials and personell?
Author Response
Manuscript ID: toxins-2573433
Title: Multi-mycotoxins analysis in Italian grains by using ultra-high-performance chromatography coupled to quadrupole orbitrap mass spectrometry
Response to Reviewer 2
Comments and Suggestions for Authors
Point 1: The authors present an important study for complex multi-toxin analysis. The study is technically state of the art and the manuscript is well written. In the Discussion additional information is necessary for the readers: Please state what steps had so far to be carried out for comparable results using standard techniques - in comparison to the results using the present combination of ultra high-performance liquid chormatorgraphy together with high-resolution mass spectrometry. Is it possible to give an overview of the resources/costs of the so far methods and the innovative multi-sample approach - with respect to instruments/materials and personell?
Response 1: Traditional methods for analyzing mycotoxins in food include thin-layer chromatography, which has low sensitivity and reproducibility and represents a useless method for studies aimed at detecting contaminants. Gas chromatography is disadvantageous because the analytes must be thermally stable and volatile, being more indicated only for the analysis of trichothecenes. High-performance liquid chromatography coupled with ultraviolet/fluorescence detectors has high sensitivity and strong specificity but is expensive and columns are usually single-target. Tandem mass spectrometry is an advantageous detector compared to ultraviolet/fluorescence detectors, as it is highly sensitive and has the ability to analyze multiple toxins at the same time, but it has low resolution and can generate false positives. Orbitrap brings advantages that cover the disadvantages of traditional methods used precisely because of the high resolution, being able to determine molecule weights and their fragments in high resolution, even in complex matrices due to their high anti-interference ability. The authors added the missing information in the manuscript.
Thank you for the time spent evaluating our manuscript.
Reviewer 3 Report
Well written and organized manuscript. Only small remarks can be made to improve the information provided.
Please, even in the introduction, specify the kind of cereal grains in which the manuscript is focused.
Line 130: “We obtained matrix effects ranging between 63 and 122%” – This matrix effect is calculated just one or for each different grain? Same question for the other validation parameters.
Line 171: Provide references for those limits recommended. Also, there are some differences in the limits depending on the cereal.
Reference 34 is no longer in place since 2021. The 2002/657/EC was replaced by the implementing regulation 808/2021.
Author Response
Manuscript ID: toxins-2573433
Title: Multi-mycotoxins analysis in Italian grains by using ultra-high-performance chromatography coupled to quadrupole orbitrap mass spectrometry
Response to Reviewer 3
Comments and Suggestions for Authors
Well written and organized manuscript. Only small remarks can be made to improve the information provided.
Point 1: Please, even in the introduction, specify the kind of cereal grains in which the manuscript is focused.
Response 1: We did study wheat. This was specified in the introduction as requested.
Point 2: Line 130: “We obtained matrix effects ranging between 63 and 122%” – This matrix effect is calculated just one or for each different grain? Same question for the other validation parameters.
Response 2: In the herein data reported, we have studied only one typology of cereal grain (wheat). All the parameters, including the matrix effect, were evaluated on a blank wheat sample. After that, the optimized method was applied to the analysis of real samples.
Point 3: Line 171: Provide references for those limits recommended. Also, there are some differences in the limits depending on the cereal.
Response 3: As suggested by reviewer 3, references have been provided.
Point 4: Reference 34 is no longer in place since 2021. The 2002/657/EC was replaced by the implementing regulation 808/2021.
Response 4: Thanks for the update, it has been adjusted here and we will take the new reference in our future studies.
Thank you for the time spent evaluating our manuscript.